# Can Health Perceptions, Credibility, and Physical Appearance of Low-Fat Foods Stimulate Buying Intentions?

**DOI:** 10.3390/foods9070866

**Published:** 2020-07-02

**Authors:** Inés Küster-Boluda, Natalia Vila

**Affiliations:** Marketing Department, University of Valencia, 46010 Valencia, Spain; natalia.vila@uv.es

**Keywords:** information credibility, physical appearance, product health perceptions, overall attitude, purchase intention, food market

## Abstract

This study examines the influence of labelling and packaging strategies on perceived product health and overall attitudes to low-fat products in the field of healthy food claims among young consumers. The aim was to determine if these aspects can influence buying intentions. After a literature review, a quantitative study was carried out. With a sample of 300 young consumers (18–25 years old) and the use of partial least square methodology, this paper demonstrated that: (1) nutritional information and visual cues affect consumers’ perceptions (information credibility and physical appearance), (2) information credibility influences product health perceptions and attitudes toward a product, (3) physical appearance affects attitudes toward a product, and (4) overall attitude to the product influences purchase intentions The results achieved show that credibility and physical appearance could stimulate low-fat foods purchase intentions through a positive global attitude to the product. Additionally, nutritional information and visual cues play a more relevant role than nutritional information response and informative cues. These results and the conclusions that follow must be understood in the analysed context (low-fat foods) with the sample used (300 young consumers).

## 1. Introduction

The packaging, among other variables, can influence individual and household food choices [1] because as cue utilization theory suggests, consumers have a propensity to use extrinsic cues as replacement indicators of product quality [2]. Cue utilization theory can explicate the increasing relevance of packaging design and strategy as a vehicle for consumer communication and branding [3]. The package turns into a primary vehicle for communication and branding [4] and a powerful weapon in the food industry [5]. This could be especially interesting when the consumption of low-fat products (a kind of functional product) is analysed among young people, because consumers are more and more worried about nutrition and food quality, even youth segments [6]. Many consumers seek safe foods that positively stimulate good health; and packaging information can help them. Generation Y/Millennials consumers (18–25 years old) are not interested in cooking, but enjoy eating, and thus depend more on packaged food. They are becoming more independent for their food and beverages choices as well as for their purchase decisions [7].

In this scene, this paper studied the perceived product health or perceived risk of disease that can be communicated through label and packaging strategies among young consumers. Due to the nature of the food advertising market, instead of advertising, companies use other methods of marketing such as packaging [5]. More specifically, firms employ packages that promote healthy consumption, because nutrition labels and packaging influence health beliefs and purchase intentions [8].

Taking this into account, this paper tried to analyse diverse young consumers’ perceptions that can affect the global attitude to low-fat products and their behaviour intention to purchase them. Specifically, a healthy food (juice with milk) versus an unhealthy one (candy bars) were considered in our research. Studies with milk packaged as a healthy product can be seen in different papers [9], and other studies [8,10] propose candies as a hedonic product. There is an expanded consumption of pre-packaged foods that usually contain high levels of sugar, fat, saturated fatty acids, trans-fatty acids, and sodium [11]. To avoid nutrition-related diseases, the World Health Organization (WHO) recommends decreasing these nutrients with the aim of increasing the nutritional value. In this regard, nutrition labelling and packaging have received considerable attention [11]. This paper argues that young consumers’ responses to labelling and packaging will affect their perceptions related to information credibility, physical appearance, and product health, as the paper explains below. It tries to answer the question: can health perceptions, credibility, and physical appearance of low-fat foods stimulate buying intentions?

Our final purpose was to provide public entities and private companies with data to increase success “in promoting healthy foods in a way that young consumers will embrace” [9].

Through this paper, we will try to fulfil diverse literature gaps. Most of the existing research on food marketing focused on television advertising only, forgetting other important marketing tools such as packaging [10,12]. Little research is available regarding consumers’ perceptions of food packaging [13], and research on nutrition information on packaged foods showed that the given information is often misinterpreted and is confusing and inappropriate for estimating an individual product’s contribution to the overall diet [11]. There seem to be relatively few consistent findings on consumers’ responses to health claims [14].

We expand the knowledge of labelling and packaging as strategic and marketing tools, especially within a competitive and mature food industry. This could be considered an extremely important research topic, because consumers place high importance on the extrinsic attributes of packaging to aid their purchase decisions [15]. In one study, over 73% of consumers agreed that they utilized packaging to assist in their purchase decisions [15]. To reach the mentioned objectives and to fulfil the literature gaps, 300 consumers between the ages of 18 to 25 years were interviewed. To test our model, partial least squares (PLS) techniques were used. Section 2 offers the theoretical foundation of this paper. Section 3 explains the methodology employed. Section 4 shows the results. Section 5 underlines the main conclusions. Although the scope of this paper excludes packaging legislation, it should not be overlooked; the two products selected follow the requirements for information provided on the packaging.

## 2. Label and Packaging Elements and Perceived Product Health

This section analyses the relationship between packaging aspects and product health perceptions.

### 2.1. How to Improve Low-Fat Food Credibility, Physical Appearance, and Product Health Perceptions

Currently, food packages include more and more information on their labels, expanding the complexity of consumer decision-making and increasing consumers’ scepticism toward food labels [16]. According to the authors of [16], it is important to evaluate the efficacy of information communicated to consumers.

Packaging elements can be classified into non-verbal or visual cues (graphics, size, colour, and shape) and verbal or informative cues (health and nutritional claims) [3,17,18,19,20]. There are some differences between them (visual vs. informative cues), as explained below.

Written cues on the package (label) could support consumers in making decisions more carefully. However, due to the excessive information shown and/or because the label presents misleading or inaccurate information, sometimes packaging information creates confusion among consumers. Manufacturers, to pack extensive information onto the label, often use very dense writing styles and very small fonts, which leads to poor readability and sometimes confusion [3]. While many consumers appreciate food labelling, they are not satisfied with standard formats; companies need to consider that consumers are paying increasing attention to label information [3].

Visual cues (graphics) comprise layout, typography, product photography, and colour combinations that build the product image [3]. A study with Chinese young consumers showed that, for them, visual packaging design of organic foods affected their preferences for this kind of food [21]. Another study confirmed that a large red heart on the front of the package or next to the menu item emphasized “heart healthiness” products [22]. Other authors [23] found that the presence of a picture on the label and package colour were the variables with the highest relative importance, regardless of consumers’ involvement with the product. “Consumers, who made their purchase on impulse tended to rely heavily on the extrinsic attributes of the packaging, especially pack photography, to assist in their choice” [24].

In summary, consumers need more useful and clear food information, and food packaging could be a good tool for this goal. The classic attitude–behaviour model [25] explains that, during their decision-making process, consumers will have different attitudes, beliefs, and behaviours. Accordingly, consumers will explore and trust different attributes or cues before deciding on their preferences of products [17,18]. In our proposal, we focused on those elements related to the label and packaging of the product (informative and visual elements). The first elements are related to the label and nutritional information on the packaging and could affect informative credibility and product health perception. The second elements are related to graphics on packaging and could influence physical appearance and product health perceptions.

The following lines argue for the possible relationship between consumers’ perceptions about the importance of nutritional information (nutritional information attitudes and responses, informative cues, and information credibility), visual aspects of the packaging (visual cues and physical appearance), and product health perceptions. Appropriate and understandable nutrition information provided on packaging affects food-choice behaviours [26], and information and visual elements can play a relevant role in this area. Figure 1 summarizes the possible relationships that arise in this paper, where the product health perceptions represent a main concept in our study.

The literature proposes that depending on whether and how consumers understand the health claim, they will build an attitude toward the claim, which in turn may affect the attitude toward the product displaying the claim [27]. A study with a sample of 1000 French consumers found a relationship between health claims and credibility [28]. Therefore, based on previous studies [28,29], nutritional and or health information context increases perceived credibility.

Other authors argue that nutrition-related claims can increase the perceived healthiness of a product [16,27,29,30,31]. Most of the research has focused on consumers at risk for chronic diseases [29]; however, attitudes and behaviour relating to nutritional information and informative cues can project a certain product health perception in consumers with or without risk of chronic diseases. The literature indicates this is true when consumers are exposed to claims that state that the product is healthy [22]. Nevertheless, it must be noted that in most studies, health labels improved perceived healthiness, but the impact was rather small [16].

Similarly, and according to previous literature [22], we expected nutritional information to have an effect of on consumers’ evaluations and purchase intentions. Positive nutritional information attitudes could lead to better product evaluations and higher purchase intentions for a packaged food product [22,32].

We made the following hypotheses where nutritional information and informative cues are related to information credibility and perceived product health.

**Hypothesis** **1** **(H1)**.
*Nutritional information attitude influences: (a) information credibility and (b) consumers’ perceptions of product health.*


**Hypothesis** **2** **(H2)**.
*Nutritional information response influences: (a) information credibility and (b) consumers’ perceptions of product health.*


**Hypothesis** **3** **(H3)**.
*Informative cues importance influences: (a) information credibility and (b) consumers’ perceptions of product health.*


Usually, consumers have the chance to examine food products before they decide to consume them [33]. In this sense, colour is one of the most relevant visual cues concerning the likely sensory properties [33] that, together with other visual cues, affect physical appearance. A review in this field [34] identifies the main effects of visual cues on consumer perceptions and behaviours. That review concludes that visual cues can affect physical appearance, product acceptance, and consumption. These results agree with others [35] that state that although fewer studies in marketing have explored product design issues, the few studies that exist show a relationship between visual indicators, physical appearance, and product acceptance.

At the same time, a large body of laboratory research has proven that making colour alterations (i.e., hue or intensity/saturation) of food and beverage items can sometimes have a dramatic impact on the expectations, and hence on the subsequent experiences of consumers [34]; one of these areas is in relation to product health perceptions.

Visual cues can affect product health perceptions because health labels are more accepted on products that already possess a healthy image, and this image can affect the perceived healthfulness of the product category [16]. In a study focused on restaurant menus, results found that visual information is more effective than providing only informational cues [36].

Visual cues could affect physical appearance and product health perceptions, as we state in the following hypothesis.

**Hypothesis** **4** **(H4)**.
*Visual cues importance influences: (a) physical appearance and (b) consumers’ perceptions of product health.*


Credibility of the message and visual appearance can affect healthiness perceptions [20]. In the food industry, and in most studies, increasing health claims of the product improves perceived healthiness [14]. Others [23] have found that lack of nutritional knowledge/credibility limits the acceptance of functional foods (product heath perceptions) and argue that the use of health claims might be necessary to assure that consumers are aware of their health benefits. In this sense, hedonic aspects such as the appearance of the packaging and health-related messages’ credibility influence product health perceptions [37].

Therefore, the increased confidence in labelling engenders stronger beliefs that the products have those attributes [20,37]. As the attribution theory states, customers evaluate whether the provided information is truthful. If source credibility is low, customers perceive information as less useful because they do not believe in it [38].

Following these premises, we propose a fifth hypothesis that states that information credibility affects product health perception.

**Hypothesis** **5** **(H5)**.
*Information credibility influences consumers’ perceptions of product health.*


On the other hand, a study of restaurants’ menus found that parents who perceive nutrition information (both numeric values and low-calorie symbols were presented) as being highly credible, perceive restaurants as healthier and have more positive perceptions overall [38]. This perceived nutrition information is related to the physical appearance perceived through visual cues. Consumers accept more health claims on products that offer a healthy visual image [14].

In this sense, the sixth hypothesis underlines that physical appearance affects product health perception.

**Hypothesis** **6** **(H6)**.
*Physical appearance influences consumers’ perceptions of product health.*


### 2.2. How to Improve Overall Low-Fat Food Attitudes in Order to Stimulate Purchase Intention

The previous sub-sections have underlined the relevance of nutritional claims in food and their effects on consumers’ evaluation of a product [39]. This point proposes that the information credibility, the physical appearance, and the health perceptions of the product can affect the global attitude toward the product. Lastly, a positive global product attitude will influence buying intention.

Related to credibility, the literature has demonstrated that information credibility plays a relevant role in the building of a positive products’ attitude [20,38]. For example, attribution theory defends that customers aim to assess whether the information delivered on a package provides accurate information and whether the source of the message is credible; if the answer is yes, consumers show a favourable attitude to the product. Another study found that participants who perceived higher levels of credibility, evaluated the packaging information more positively and were more likely to act based on this information [40].

Therefore, we developed the following hypothesis that defends an influence of information credibility on the global attitude of the product.

**Hypothesis** **7** **(H7)**.
*Information credibility influences the global attitude to the product.*


Regarding physical appearance, the literature states that the way the food is exposed can affect attitudes to the food, because the acceptance of a novel food will be better if the consumer has visual exposure [34]. Visual exposure facilitates the acceptance and attitude of a food in children that are exposed to novel food pictures, more than those not visually exposed [34]. As stated before, the literature [20,38] finds that physical appearance of a restaurant menu and positive attitudes to it are related if both numeric values and low-calorie symbols are presented.

So, we can hypothesize that physical appearance can influence the overall product attitude.

**Hypothesis** **8** **(H8)**.
*Physical appearance influences the global attitude to the product.*


Relating to product health perceptions, in defining consumers’ foods acceptance, consumers’ perceptions of the healthiness of products are essential [23]. However, healthiness of foods influences food attitude and, also, the amount of the food intake [41]. Based on the literature review, we present a ninth hypothesis that states that product health perceptions affect overall attitude to the product.

**Hypothesis** **9** **(H9)**.
*Product health perception influences the global attitude to the product.*


Finally, we propose a possible link between an overall attitude to the product and purchase intention. In addition to the arguments presented above, this relationship appears to be implicit in the works of other authors [19,20]. For example, a review in this field [34] concludes that if a consumer shows a positive attitude to a product, the intention to try it will increase; that is, consumers who showed higher levels of perceived credibility, assessed the information more favourably and were more likely to act based on this information [40]. In summary, attitudes may affect purchase intentions and, ultimately, purchasing behaviour [27]. So, a last hypothesis states that an overall attitude to the product affects its purchase intention

**Hypothesis** **10** **(H10)**.
*The global attitude to the product influences the purchase intention.*


## 3. Methodology

### 3.1. The Population

This research was developed in Spain, a developed country in the south of the UE. More specifically, it was carried out in a Spanish region, Valencia, with a similar profile to Spain overall [42]. This paper focused on the youth market, because, as justified in the Introduction Section, this segment appears to be a relevant field for research, since adolescents are more often away from home and the watchful eyes of their parents [18]. Adolescents have particular food choices and meal habits different from those of younger children and adults [20]

As Table 1 illustrates (*n* = 300 consumers from 18 to 25 years old), 53% of the respondents were men and 47% were women. Most of the sample (78.3%) were single, had a university education or more (75%), and their incomes were under the mean or far under the mean (63%).

### 3.2. Information Collection

Before the quantitative study, a qualitative preliminary one (focus group) was carried out based on seven young Spanish consumers’ opinions. It was done to obtain feedback on the suitability of the measuring instrument and to determine the products selected. This qualitative phase was useful to refine the final questionnaire, because the help of the students was key to writing the items of the questionnaire properly

After that, the quantitative study started in July 2017. Samples were obtained personally by an external company at the door of five different faculties in a university campus and at a public hospital. This campus was chosen because in addition to the faculties, there was a public hospital, where people from different educational levels, not only university students, attended [17,18].

First, we tested that participants had bought the products considered in our study (candies or juice with milk, Figure 2), and, after this, they were requested to participate in our research by completing a structured and personal questionnaire. A final valid sample of 300 participants was collected (error = 5.66 if sample procedure was probabilistic). This sample size must be considered when generalizing results.

### 3.3. Measurement and Techniques

As Appendix A shows, and based on the literature, diverse multiple-item Likert-type scales were used in the present study to measure the following concepts: NIA: nutritional information attitude, NIR: nutritional information response, IC: label/informative importance, VC: packaging/visual importance, C: information credibility, PHP: perceived product health/perceived risks of disease, PA: physical appearance, GA: global attitude to the product, and PI: Purchase intention.

PLS (partial least squares) was employed because it avoids normal distribution of the data and can simultaneously evaluate the measurement and the theoretical models. Additionally, this technique simultaneously approximates all path coefficients and individual item loadings in the context of a specified model and, as a result, allows eluding biased and inconsistent parameter estimations [43]. Furthermore, PLS allows for the assessment of interactions by reducing type II errors [44]. Therefore, prior to the hypotheses test, the measurement instrument was analysed, following the same procedure used in previous studies [43,45]; that is, psychometrics characteristics (reliability, validity, and accuracy of the estimation) of the measurement model were verified, as explained below.

Through the evaluation of the reliability of the individual items and the discriminant validity of the constructs, we analysed the adequacy of the measures [46]. By exploring the loading of the measures on their corresponding construct, item reliability was calculated. All the loadings of scales are proximate or exceeded 0.6 [47]. Composite reliability was used to analyse the reliability of the constructs, since this has been regarded as a more exacting measurement than Cronbach’s alpha [48]. Because the composite reliability values exceeded the threshold of 0.7, all constructs were reliable [49].

The measures showed convergent validity because the AVE (average variance of manifest variables) was at least 0.5. The square root of AVE was used to evaluated discriminant validity, which should be greater than the correlation between the construct and other constructs in the model [50]. All variables had discriminant validity. In all cases, the HTMT (heterotrait-monotrait ratios) surpassed the value 0.85, confirming discriminant validity.

The blindfolding technique was used to calculate the Q2 and the R2 to study the accuracy of the estimation. As all values of Q2 showed positive values, the relations in the model had predictive relevance. Just one construct, physical appearance (AF), presented an R2 < 0.01. In this sense, the hypotheses associated with this variable show a low predictive power.

## 4. Results and Discussion

Prior to the hypotheses test, and with the aim of better understanding the results, we carried out a descriptive analysis.

After the scales psychometrics characteristics analysis, structural equation models through PLS were used to test the hypotheses. For these tests, the bootstrapping approach was used (sets of 5000 samples were formed to obtain 5000 estimates for each parameter in the model, and each sample was acquired by sampling with a replacement of the original data set) [45,48].

In the theoretical model, seven relationships were significant, and the model demonstrated a good fit and appropriated R^2^ values (Table 2).

As Table 2 shows, first, nutritional information attitude (H1a) and informative cues (H3a) have a significant influence on information credibility. In this sense, aspects such as the utility of the nutrition information, the health slogans, the readability, or the label design have a significant effect on the perceived credibility of the information to the consumers. These aspects help the brand to be perceived as serious, trustworthy, and honest. These results are in line with previous studies [28,29].

Second, and according to previous results [34], the visual cues’ importance stimulates the physical appearance of the product (H4a). Features such as the colours, materials, and the use of images positively affect the physical appearance of the product.

Third, information credibility influences significantly on consumers’ perceptions of product health (H5). As attribution theory states [38], if a consumer considers that the information on the label/package is true, serious, and honest, he/she will perceive the product as healthier than if he/she considers that the label/packaging information is not credible. Other authors reach similar results [14,37].

Fourth, information credibility (H7) and physical appearance (H8) significantly influence the overall attitude to the product. Both visual and informative characteristics of the packaging affect consumers’ attitudes to the product, as previous studies suggest [34,38,40].

Fifth, there is a significant influence of the global attitude to the product on the purchase intention (H10). Usually, a consumer who shows a positive attitude to a product is more likely to buy it [34].

Finally, not significant relationships (the rest of the hypotheses) were found between some established concepts (nutritional information attitude and consumers’ perceptions of product health; nutritional information response and information credibility; nutritional information response and product health consumers’ perceptions; informative cues importance and consumers’ perceptions of product health; visual cues importance and consumers’ perceptions of product health; physical appearance influences consumers’ perceptions of product health and product health perception and global attitude to the product). Some of these results can be explained because of the cultural differences among countries [29] or by the market segment analysed.

In response to the question formulated in the title of this paper (can health perceptions, credibility, and physical appearance of low-fat foods stimulate buying intentions?) and taking into account the characteristics of the sample analysed, the results achieved allow us to state that credibility and physical appearance can stimulate low-fat foods purchase intentions through a positive global attitude to the product. Additionally, nutritional information and visual cues play a more relevant role than nutritional information response and informative cues.

## 5. Implications, Limitations, and Further Research

The World Health Organization (WHO) recommends that institutions and firms need to think about strategies that provide balanced information for consumers to help them to make healthy food choices easily, with an emphasis on accurate, standardized, and comprehensible information on food labels [29]. In this sense, this paper analysed nutrition information that can be communicated by the food label and by nutrition-related claims on the package, which can help consumers interpret the Nutritional Facts table values. With a sample of 300 young consumers of candies and/or juice with milk, our research demonstrated that nutritional information and visual aspects of the label/packaging affect consumers’ perceptions and attitudes to a product, and these influences would affect future purchases. However, the small sample size must be considered when generalizing to other contexts.

In relation to managerial implications and following the WHO recommendations [51], we suggest some strategies and actions for companies or public institutions interested in launching a healthy product in the market.

First, institutions and firms must consider the importance of information credibility and its effects on product health perceptions and global attitude to the product. Not in vain, our results suggest that managers should stress the healthy message of their product to reinforce their credibility. As mentioned above, packaging strategies should consider the use of the utility of the nutrition information, the health slogans, the readability, and the label design.

Second, together with attention to the aspects that may have an impact on information credibility, those responsible for packaging should focus their efforts on physical appearance. Not surprisingly, the results show the influence of physical appearance on the global attitude toward the product.

Third, and because packaging cues (information and visual elements) play a significant role on shaping a positive global attitude to the product, managers should not neglect both aspects. Claims must go together with a suitable label and packaging design. In this sense, they must combine both features to try to maximize the effects of packaging design on consumer behaviours. Results show that a global attitude to the product influences purchase intention [22].

And lastly, as explained before, our results do not support previous relationships found in the literature. There are cultural differences between consumers in different countries that could explain these results. In this sense, we recommend managers to consider these differences between countries [29]. This is not the case in some firms that did not hear their consumers (i.e., Carlsberg in the Danish market or Kellogg in the Indian market) [52]. This is especially true when we consider food products and healthy claims [31].

In summary, our research proposes the following recommendations to food managers who desire to launch a low-fat food product to the market:-Managers must focus on consumers’ nutritional attitudes and informative cues in order to improve information credibility.-If managers work to achieve this credibility, they will transmit good product health perceptions to their customers.-Managers should not ignore aspects related to the visual elements of the package as the way to develop an attractive physical appearance for their customers.-Managers who wish to improve low-fat food global attitudes to stimulate purchase intention should focus on information credibility and physical appearance.-Finally, and to stimulate low-fat foods sales, managers should generate a positive overall attitude to the product. It could be interesting to develop strategies that concentrate on the development of good feelings in the target market.

Finally, our study is not without some limitations that represent possible further research lines. The main limitation is related to the sample (300 young consumers). Although other studies [7] consider similar sample sizes and composition, the conclusions and implications must be considered with caution, and further research could expand the sample. Additionally, some moderations of variables such as the product category or the claim design could be studied to a better understand the results. In addition, some more packaging design variables could be considered to design “the best packaging” in terms of sales. Other targets, apart from young consumers, could be analysed, such as adolescents. Packaging offers a wide field of study where researchers from diverse scopes can collaborate. For example, further research could study consumers’ perceptions to active food packaging [53] or to smart packaging systems [54]. Last but not least, the study could be extended to other consumer segments, for example, the elderly.

## Figures and Tables

**Figure 1 foods-09-00866-f001:**
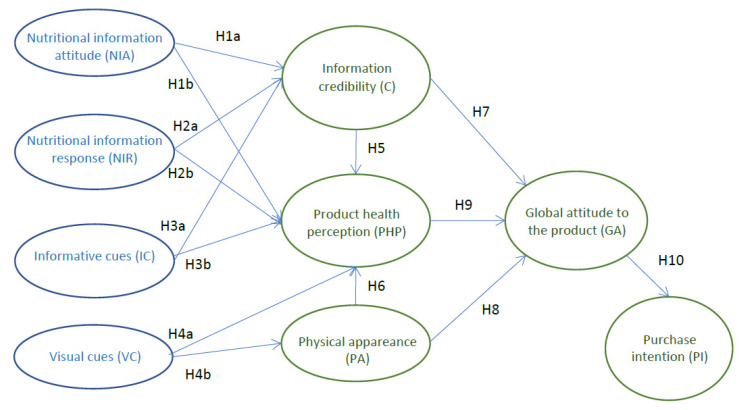
Theoretical framework.

**Figure 2 foods-09-00866-f002:**
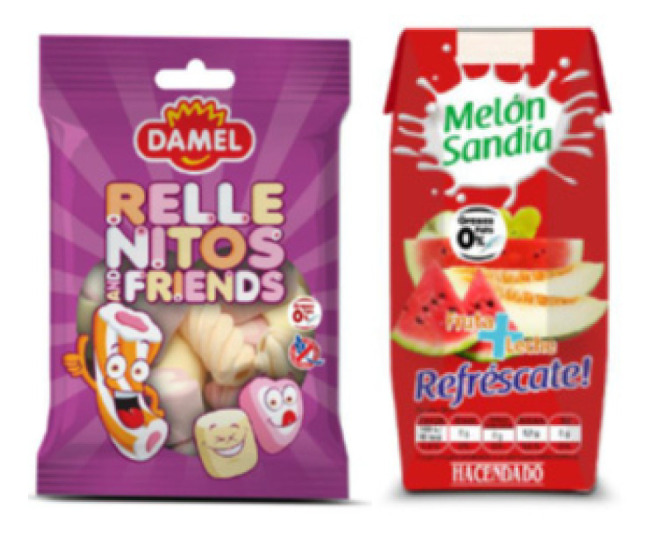
Products considered.

**Table 1 foods-09-00866-t001:** Sample description.

Variable	%	Variable	%	Variable	%
**Gender**		**Studies**		**Incomes** (ref. 900e)	
Male	53%	Elemental	2.7%	Far under the mean	43.0%
Female	47%	High school	22.3%	Under the mean	20.0%
**Civil status**		University	65.3%	In the mean	20.0%
Single	78.3%	Master/PhD	9.7%	Above the mean	14.3%
Married (in couple)	21.7%			Well above the mean	2.7%

Note: *n* = 300; Age mean: 21.82 years old.

**Table 2 foods-09-00866-t002:** Hypotheses testing.

Hypotheses	Stand. Coef	T Value	Contrast
**How to improve low-fat food credibility, physical appearance, and product health perceptions**			
**H1a: Nutritional inf. attitude** **➔** **Information credibility**	**0.270**	**5.310 *****	**Yes**
H1b: Nutritional inf. attitude ➔ Product health	−0.013	0.230 ns	No
H2a: Nutritional inf. response ➔ Information credibility	0.030	0.424 ns	No
H2b: Nutritional inf. response ➔ Product health	−0.004	0.042 ns	No
**H3a: Informative cues** **➔** **Information credibility**	**0.123**	**1.952 ***	**Yes**
H3b: Informative cues ➔ Product health	0.027	0.336 ns	No
**H4a: Visual cues** **➔** **Physical appearance**	**0.148**	**1.710 ***	**Yes**
H4b: Visual cues ➔ Product health	0.020	0.247 ns	No
**H5: Information credibility** **➔** **Product health**	**−0.282**	**4.368 *****	**Yes**
H6: Physical appearance ➔ Product health	−0.061	0.829 ns	No
**How to improve low-fat food global attitude to stimulate purchase intention**			
**H7: Information credibility** **➔** **Overall attitude to product**	**0.422**	**7.658 *****	**Yes**
**H8: Physical appearance** **➔** **Overall attitude to product**	**0.318**	**5.561 *****	**Yes**
H9: Product health ➔ Overall attitude to product	−0.049	1.087 ns	No
**H10: Overall attitude to product** **➔** **Purchase intention**	**0.643**	**20.373 *****	**Yes**

* *p* < 0.1; *** *p* < 0.01. The bold letters to underline the hypotheses supported.

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
