# Peer review of "Can Health Perceptions, Credibility, and Physical Appearance of Low-Fat Foods Stimulate Buying Intentions?"

_foods, 2020, doi:10.3390/foods9070866_

Round 1

Reviewer 1 Report

General comments

The manuscript did not take into account the requirements legislation puts to the information to be provided in the packaging.

A sample of 300 participants. Is this enough to draw conclusions? The authors should elaborate on this.

Overall the writing style is missing scientific rigor and the red thread is sometimes missing. This would require that the authors would review their manuscript completely: questioning what is important and what is not, what is the best order of presenting things, what wording is best to use and where visual aids are needed.

Professional help is needed with the language.

Abstract

The abstract requires rewriting. It has round statements: (paper  offers some interesting conclusions…) It does not reveal, how or towards what direction attitudes or buying habits are shift. Two last sentences need complete rewriting to be understandable, and valuable for the reader, and correct English.

Introduction

Some sentences fail to convey the message they suppose to. Examples:

" Individual and household food choices are a function of numerous influences": Meaning? The first sentence of the manuscript should be strong, not vague like this.

“ Companies and institutions should know if their products’ labels and packaging reduce the perceived risk of disease" : Do labels themselves really reduce the risk of disease?

In addition, the Introduction would benefit from rearranging topics introduced, starting from a wide viewpoint, and narrowing it gradually to this research.

Chapter 2

The first paragraph seems to belong to the Introduction

Remove/revise unnecessary phrases like “as the paper explains below”, “in the same line”, “in this sense”

There is no text under headline 2.1.

It is hard to follow the logic of the hypotheses. Could this be presented with a figure that presents building the hypotheses based on a series of steps, for example in a style of a decision making tree? The end result would be Fig.1 type of solution (which by the way is not well explained to the reader) but the road leading to the figure would benefit from logical and visual reasoning.

References marked sometimes with two referencing techniques, both Harvard and numbering.

Chapter 3

The sample size should be mentioned at the start of this chapter.

“Therefore, prior to the  hypotheses test, the measurement instrument was analyzed, following the same procedure used in previous studies [40, 42]”: Please elaborate how, not just reference to previous studies.

Chapter 4

It goes beyond my capabilities to assess the result, and how they were derived and analyzed. But the analysis might use more elaboration on how different aspects influence on others and why.

Author Response

Dear reviewer,

Thank you so much for your time and effort. We are extremely grateful for the interesting and useful comments you sent to us regarding improvements to our manuscript. We really appreciate it. As a result of your suggestions, we have made the following changes. Following the journal recommendation, we have used the "Track Changes" function in Microsoft Word, so that changes are easily visible to the editors and reviewers.

Point 1: The manuscript did not take into account the requirements legislation puts to the information to be provided in the packaging.

Response 1: The aim of this paper does not focus on legislation requirements. In any case, the two products (candy and juice with milk) follow the legislation.

Point 2: A sample of 300 participants. Is this enough to draw conclusions? The authors should elaborate on this.

Response 2: According to the reviewer, the sample size could limit the generalizability of the results, but other studies in larger populations, use similar sample sizes (i.e. Kapoor & Kumar, 2019; Lin, 2019; Lorestani & Khalili, 2019). In any case, we have added an explanation in methodology and conclusions sections. Additionally, the sample error (if the sample procedure was probabilistic) has been calculated (error=5.66). For these reasons, we consider that this sample size is enough.

Point 3: Overall the writing style is missing scientific rigor and the red thread is sometimes missing. This would require that the authors would review their manuscript completely: questioning what is important and what is not, what is the best order of presenting things, what wording is best to use and where visual aids are needed.

Response 3: Following your recommendations, all the paper has been read and rewritten.

Point 4: Professional help is needed with the language.

Response 4: We really apologize for your perception. We used a professional service (Proof-Reading-Service.com) and we have the certificate of their review. To improve the quality, we have sent again to the company.

Point 5: Abstract: The abstract requires rewriting. It has round statements: (paper offers some interesting conclusions…) It does not reveal, how or towards what direction attitudes or buying habits are shift. Two last sentences need complete rewriting to be understandable, and valuable for the reader, and correct English.

Response 5: Following your comments, the abstract has been rewritten.

Point 6: Introduction: Some sentences fail to convey the message they suppose to. Examples:

" Individual and household food choices are a function of numerous influences": Meaning? The first sentence of the manuscript should be strong, not vague like this.

Companies and institutions should know if their products’ labels and packaging reduce the perceived risk of disease" : Do labels themselves really reduce the risk of disease?

Response 6: According to the reviewer, Introduction has been rewritten, deleting some sentences, and reinforcing the message.

Point 7: Introduction: In addition, the Introduction would benefit from rearranging topics introduced, starting from a wide viewpoint, and narrowing it gradually to this research.

Response 7: We have tried to modify the argumentation.

Point 8: Chapter 2: The first paragraph seems to belong to the Introduction

Response 8: We have moved that paragraph to the Introduction

Point 9: Remove/revise unnecessary phrases like “as the paper explains below”, “in the same line”, “in this sense”

Response 9: Where possible, in most cases, these phrases have been deleted.

Point 10: There is no text under headline 2.1.

Response 10: This headline has been deleted.

Point 11: It is hard to follow the logic of the hypotheses. Could this be presented with a figure that presents building the hypotheses based on a series of steps, for example in a style of a decision making tree? The end result would be Fig.1 type of solution (which by the way is not well explained to the reader) but the road leading to the figure would benefit from logical and visual reasoning.

Response 11: To a better understanding of the logic of the hypotheses, a graphical representation of the paper has been added just alongside with the abstract, after the keywords. Additionally, Figure 1 has been moved at the beginning of hypotheses justification.

Point 12: References marked sometimes with two referencing techniques, both Harvard and numbering.

Response 12: We have tried to homogenize references, using numbering technique as the journal requests.

Point 13: Chapter 3: The sample size should be mentioned at the start of this chapter.

Response 13: We have added the sample size at the beginning of Section 3.

Point 14: Chapter 4: “Therefore, prior to the hypotheses test, the measurement instrument was analyzed, following the same procedure used in previous studies [40, 42]”: Please elaborate how, not just reference to previous studies.

Response 14: We have added a sentence to clarify this statement. And the procedure is explained in the following lines.

Point 15: It goes beyond my capabilities to assess the result, and how they were derived and analyzed. But the analysis might use more elaboration on how different aspects influence on others and why.

Response 15: To clarify results interpretation and following other reviewer’s suggestion, Section 4 and Section 5 have been rewritten.

Thank you again for your time and effort. We hope this new version has answered your questions and suggestions for improvement.

Reviewer 2 Report

Healthy food consumption has become a topic of increasing interest over the last decades, both for academicians and practitioners. Given the importance of the subject, any effort of improving knowledge on this topic represents a timely research idea, especially to bring order in the existing literature and to provide new suggestions for food companies. Some concerns with the study do remain which require authors’ revisions.

INTRODUCTION. Overall, the Introduction is well written. However it could be improved as follows. First, the aim of the paper should be well declined, as written on pag.2 (lines 71-73). Second, a structure of the paper is usually presented at the end of the Introduction section to enhance the paper readability.

LITERATURE. The paper cites an appropriate range of literature source, which supports the research hypotheses’ development. However, most of the literature dates back to before 2015, with many even older references. A careful revision of recent literature is recommended to improve the timeliness of the current research. Moreover, it would be appropriate to consider the breadth of the theoretical debate on the multiple functions that packaging could perform to improve food safety, by considering, for example the topics of packaging biodegradability/sustainability [e.g. Atarés, L., & Chiralt, A. (2016). Essential oils as additives in biodegradable films and coatings for active food packaging. Trends in food science & technology48, 51-62], the technical aspects of the packaging [e.g. Malhotra, B., Keshwani, A., & Kharkwal, H. (2015). Antimicrobial food packaging: potential and pitfalls. Frontiers in microbiology6, 611] as well as the use of intelligent packaging aimed at satisfying the demand for safer foods with better shelf life [e.g. Biji, K. B., Ravishankar, C. N., Mohan, C. O., & Gopal, T. S. (2015). Smart packaging systems for food applications: a review. Journal of food science and technology52(10), 6125-6135].

METHODOLOGY. The sample includes only young people aged between 18-25 years. More explanations are needed to justify your sample choice. Moreover, it would be better to move the sample description (sub-section 4.1) in the methodology section, after the description of the population. My further doubt concerns the sample representativeness, since the study focused on 300 respondents. Is this number statistically significant with respect the Spanish population considered? You should provide more in-depth explanations about your sample size and composition.

CONCLUSION. Actually, theoretical implications aren't argued. In this section, Authors just repeat a summary of the results without discussing them in light of previous literature. Theorethical implications should debate the research findings comparing them with other studies, including several citations and references. They should highlight how the present research contributes to the extant literature, thus supporting its relevance in the scientific debate. A separate section with conclusions, limits and future research is suggested. Please, add methodology concerns (i.e. sample size and composition) in your limitations.

MINOR REVISION. The language is accurate, which enhances the paper readability. However, a careful revision of the paper is recommended to eliminate some refusals (e.g.: on p.2, line 67 there is a dot instead of a comma). On pag.1, line 27 you write “low-sugar”: I suggest to be more generic by including also low-fat, low sodium, nutritional values, etc.

Author Response

Dear reviewer,

Thank you so much for your time and effort. We are extremely grateful for the interesting and useful comments you sent to us regarding improvements to our manuscript. We really appreciate it. As a result of your suggestions, we have made the following changes. Following the journal recommendation, we have used the "Track Changes" function in Microsoft Word, so that changes are easily visible to the editors and reviewers.

Point 1: INTRODUCTION. Overall, the Introduction is well written. However it could be improved as follows. First, the aim of the paper should be well declined, as written on pag.2 (lines 71-73). Second, a structure of the paper is usually presented at the end of the Introduction section to enhance the paper readability.

Response 1: Following your recommendation, Introduction has been changed, reinforcing the aim of the paper and introducing its structure.

Point 2: LITERATURE. The paper cites an appropriate range of literature source, which supports the research hypotheses’ development. However, most of the literature dates back to before 2015, with many even older references. A careful revision of recent literature is recommended to improve the timeliness of the current research. Moreover, it would be appropriate to consider the breadth of the theoretical debate on the multiple functions that packaging could perform to improve food safety, by considering, for example the topics of packaging biodegradability/sustainability [e.g. Atarés, L., & Chiralt, A. (2016). Essential oils as additives in biodegradable films and coatings for active food packaging. Trends in food science & technology48, 51-62], the technical aspects of the packaging [e.g. Malhotra, B., Keshwani, A., & Kharkwal, H. (2015). Antimicrobial food packaging: potential and pitfalls. Frontiers in microbiology6, 611] as well as the use of intelligent packaging aimed at satisfying the demand for safer foods with better shelf life [e.g. Biji, K. B., Ravishankar, C. N., Mohan, C. O., & Gopal, T. S. (2015). Smart packaging systems for food applications: a review. Journal of food science and technology52(10), 6125-6135].

Response 2: According to the reviewer suggestion, some more recent papers have been incorporated. We really appreciated the studies you recommend to us, but we consider that they are out of our field. In any case, we have added some of them as further research lines.

Point 3: METHODOLOGY. The sample includes only young people aged between 18-25 years. More explanations are needed to justify your sample choice. Moreover, it would be better to move the sample description (sub-section 4.1) in the methodology section, after the description of the population. My further doubt concerns the sample representativeness, since the study focused on 300 respondents. Is this number statistically significant with respect the Spanish population considered? You should provide more in-depth explanations about your sample size and composition.

Response 3: Following your comments: (1) young people choice have been justified and introduced in the text, (2) the sample description has moved to Section 3. Additionally, and according to the reviewer, the sample size could limit the generalizability of the results, but other studies in larger populations, use similar sample sizes (i.e. Kapoor & Kumar, 2019; Lin, 2019; Lorestani & Khalili, 2019). In any case, we have added an explanation in methodology and conclusions sections. Additionally, the sample error (if the sample procedure was probabilistic) has been calculated (error=5.66). For these reasons, we consider that this sample size is enough.

Point 4: CONCLUSION. Actually, theoretical implications aren't argued. In this section, Authors just repeat a summary of the results without discussing them in light of previous literature. Theorethical implications should debate the research findings comparing them with other studies, including several citations and references. They should highlight how the present research contributes to the extant literature, thus supporting its relevance in the scientific debate. A separate section with conclusions, limits and future research is suggested. Please, add methodology concerns (i.e. sample size and composition) in your limitations.

Response 4: Following your recommendation and from other reviewer, Section 4 and Section 5 have been rewritten. And according to the editor and journal recommendations, Section 5 has been trimmed (the journal suggests no more than half-page). Additionally, methodology concerns have been stated.

Point 5: MINOR REVISION. The language is accurate, which enhances the paper readability. However, a careful revision of the paper is recommended to eliminate some refusals (e.g.: on p.2, line 67 there is a dot instead of a comma). On pag.1, line 27 you write “low-sugar”: I suggest to be more generic by including also low-fat, low sodium, nutritional values, etc.

Response 5: We have reviewed the spelling and grammar, trying to eliminate the possible mistakes.

Thank you again for your time and effort. We hope this new version has answered your questions and suggestions for improvement.

Reviewer 3 Report

Dear Authors,

your paper entitled “Can Health Perceptions, Credibility and Physical Appearence of Low-Fat Foods Stimulate Buying Intentions?” is interesting and quite original. The presented findings are interesting and well described. However, in my humble opinion, it has substantial lacks and, although it could improve the knowledge on topic, the overall construction of the paper must be reviewed and several modifications and integrations are needed.

Please, see following suggestions.

  • Literature review dedicated to the topic is not sufficient to support findings. Furthermore, your sample defines a specific young generation (Millennials or Gen Z? See below) but you do not report any papers on “Millennials and/or GenZ’s” behavior and perception. Please, improve the literature review and the used references list. Therefore, I would recommend taking in consideration other papers dedicated to the topic of paper improving literature review
  • This paragraph is very lacking in term of information as to sample and related survey (3.1). It is not clear when the survey was carried out and the method used e.g. questionnaire. If any, the questionnaire design is an important phase of survey: its brief description should be reported in Methodology paragraph and the last version should be inserted in appendix improving the transparency on activities made. Moreover, the population selection is not supported by references and, in my opinion, can be defined as Gen Z (or Millennials, if the survey carried out before 2016). Moreover, Information collection (3.2) indicates a qualitative assessment by seven young consumers. This is a focus group or a pre-test of questionnaire? Lastly, 300 young consumers were interviewed, but how? Why did you choice university campus and one hospital to define the sample?
  • "Findings and Discussion" paragraph do not contain the discussion that in my opinion is presented in conclusion paragraph.
  • In my opinion, Conclusion paragraph should be separated from Discussion contents, therefore, I would suggest to write a new paragraph dedicated to the discussion of results in accordance to integrated literature review.

Author Response

Dear reviewer,

Thank you so much for your time and effort. We are extremely grateful for the interesting and useful comments you sent to us regarding improvements to our manuscript. We really appreciate it. As a result of your suggestions, we have made the following changes. Following the journal recommendation, we have used the "Track Changes" function in Microsoft Word, so that changes are easily visible to the editors and reviewers.

Point 1: Literature review dedicated to the topic is not sufficient to support findings. Furthermore, your sample defines a specific young generation (Millennials or Gen Z? See below) but you do not report any papers on “Millennials and/or GenZ’s” behavior and perception. Please, improve the literature review and the used references list. Therefore, I would recommend taking in consideration other papers dedicated to the topic of paper improving literature review.

Response 1: Following your suggestion, we have tried to improve our literature review, adding new references about Millennials and/or GenZ’s generations.

Point 2: This paragraph is very lacking in term of information as to sample and related survey (3.1). It is not clear when the survey was carried out and the method used e.g. questionnaire. If any, the questionnaire design is an important phase of survey: its brief description should be reported in Methodology paragraph and the last version should be inserted in appendix improving the transparency on activities made. Moreover, the population selection is not supported by references and, in my opinion, can be defined as Gen Z (or Millennials, if the survey carried out before 2016). Moreover, Information collection (3.2) indicates a qualitative assessment by seven young consumers. This is a focus group or a pre-test of questionnaire? Lastly, 300 young consumers were interviewed, but how? Why did you choice university campus and one hospital to define the sample?

Response 2: According to review suggestions, Section 3 has introduced additional information.

Point 3: "Findings and Discussion" paragraph do not contain the discussion that in my opinion is presented in conclusion paragraph.

Response 3: Following your recommendation and those of another reviewer Section 4 and Section 5 have been rewritten.

Point 4: In my opinion, Conclusion paragraph should be separated from Discussion contents, therefore, I would suggest to write a new paragraph dedicated to the discussion of results in accordance to integrated literature review.

Response 4: As stated before, Section 4 and Section 5 have been rewritten.

Thank you again for your time and effort. We hope this new version has answered your questions and suggestions for improvement.

Round 2

Reviewer 1 Report

1) Regarding the first point in the 1st round of review. The scope might not be in packaging legislation, but it should not be overlooked, as legislation sets the minimum requirements for information provided in the packaging.

2) It would make the research more tangible if the two food products or their packages were illustrated in the manuscript.  You could perhaps cover the brand name if needed.

3) Limitations of this study are not clearly discussed

4) Please check, that the question you pose in the title is answered in the article, starting from the abstract. In addition, what are the attributes that "does not matter" or do not influence the purchasing decision?

5) The language has improved since the last read, and mostly it is mostly because of removing unnecessary phrases. Some proofreading services just check the grammar, without paying attention to the flow or the meaning of the sentence.

That being said, there are still problems, especially with the new sentences, e.g.:

"The reason to choose this campus is because there is a university 450 hospital where participants from diverse educational ranges could participate [17]."

Author Response

Dear reviewer,

Thank you again for this second review. Considering your suggestions, we have made the following changes. As in the first review, we have used the "Track Changes" function in Microsoft Word, so that changes are easily visible to the editors and reviewers. The changes in the first review are in red.

Point 1: Regarding the first point in the 1st round of review. The scope might not be in packaging legislation, but it should not be overlooked, as legislation sets the minimum requirements for information provided in the packaging.

Response 1: We appreciate your comment. In this sense, it has been added a sentence related to this issue.

Point 2: It would make the research more tangible if the two food products or their packages were illustrated in the manuscript.  You could perhaps cover the brand name if needed.

Response 2: Following your suggestion, we have added the two food products considered in our research (Figure 2).

Point 3: Limitations of this study are not clearly discussed

Response 3: Section 5 has tried to discuss study limitations. We would like to notice that the journal recommended us that this section should not go beyond half a page.

Point 4: Please check, that the question you pose in the title is answered in the article, starting from the abstract. In addition, what are the attributes that "does not matter" or do not influence the purchasing decision?

Response 4: At your suggestion, we have tried to answer the question.

Point 5: The language has improved since the last read, and mostly it is mostly because of removing unnecessary phrases. Some proofreading services just check the grammar, without paying attention to the flow or the meaning of the sentence.

That being said, there are still problems, especially with the new sentences, e.g.: "The reason to choose this campus is because there is a university 450 hospital where participants from diverse educational ranges could participate [17]."

Response 5: We must apologize again for the inconvenience caused. The document has been revised again

Thank you again for your time and effort. We hope this new version has answered your questions and suggestions for improvement.

Reviewer 3 Report

Dear Author(s),

Thank you for taking into account my suggestions and carried out some modifications I asked for. In this new version, the manuscript includes integrations that I appreciated. However, I believe that it is not enough. Some integrations should be made i.e.

  1. It is not clear when the survey was carried out and the method used for the investigation e.g. questionnaire (online, PAPI …), semi-structured interview etc. Please provide time period of the survey. This is necessary to define literature review and sample.
  2. Your pilot test for the questionnaire was carried out by only seven consumers (students?!). This is the size for a focus group not for a pre-test sample. Please, justify.
  3. Moreover, you have not inserted in attachment the questionnaire and/or related track.
  4. If I understand, the sample was composed by University students. In this case, “young consumers” should be changed in “university students”. In my opinion, university students are a specific target with particular characteristics and it is not necessarily similar to “young consumers”.
  5. Some sentences seem to be extracted from "Health/Nutrition food claims and low-fat food purchase: Projected personality influence in young consumers" (Journal of Functional Foods, 2017) but this paper is not cited and inserted in the references list. Please, justify.
  6. You deleted "Vila, N.; Küster, I. (2016): “Adolescents' food  packaging perceptions. Does gender matter when weight control and health motivations are considered?” as citation but in my opinion it was useful to strength the literature review and discussion.
  7. I would suggest to re-write the abstract in order to define more in-depth the contents of the paper.

Author Response

Dear reviewer,

Thank you again for this second review. Considering your suggestions, we have made the following changes. As in the first review, we have used the "Track Changes" function in Microsoft Word, so that changes are easily visible to the editors and reviewers. The changes in the first review are in red.

Point 1: It is not clear when the survey was carried out and the method used for the investigation e.g. questionnaire (online, PAPI …), semi-structured interview etc. Please provide time period of the survey. This is necessary to define literature review and sample.

Response 1: Following your comment, the information has been added in Section 3.

Point 2: Your pilot test for the questionnaire was carried out by only seven consumers (students?!). This is the size for a focus group not for a pre-test sample. Please, justify.

Response 2: According to review suggestions, Section 3 has introduced additional information, because as the reviewer indicates, the preliminary phase to refine the final questionnaire was carried out through a focus group with seven students. Their help was key to define the items of the questionnaire properly. So, we have clarified this point in Methodology section.

Point 3: I would suggest to re-write the abstract in order to define more in-depth the contents of the paper

Response 3: Following your suggestion, we have rewritten the abstract.

Point 4: Moreover, you have not inserted in attachment the questionnaire and/or related track.

Response 4: We are sorry, but we have not inserted the questionnaire because the Appendix shows the scales used to measure the concepts. If you consider we must add it, we can create a link to access to it.

Point 5: If I understand, the sample was composed by University students. In this case, “young consumers” should be changed in “university students”. In my opinion, university students are a specific target with particular characteristics and it is not necessarily similar to “young consumers”.

Response 5: The young people include not only university students. We have clarified that the campus has been chosen because, alongside the faculties, there is a public university hospital, where people from different educational levels, not only university students, attend.

Point 5: Some sentences seem to be extracted from "Health/Nutrition food claims and low-fat food purchase: Projected personality influence in young consumers" (Journal of Functional Foods, 2017) but this paper is not cited and inserted in the references list. Please, justify.

Response 5: We have added this paper.

Point 6: You deleted "Vila, N.; Küster, I. (2016): “Adolescents' food packaging perceptions. Does gender matter when weight control and health motivations are considered?” as citation but in my opinion it was useful to strength the literature review and discussion.

Response 6: We had deleted this paper because we have added a most recent one (Küster, I., Vila, N., & Sarabia, F. (2019). Food packaging cues as vehicles of healthy information: Visions of millennials (early adults and adolescents). Food research international, 119, 170-176). But, following your recommendation, we have added again the paper.

Thank you again for your time and effort. We hope this new version has answered your questions and suggestions for improvement.
